# Metabolome and Transcriptome Reveal the Regulatory Mechanism of Anthocyanin Synthesis in Tuber Skin and Flesh of *Dioscorea alata* L.

**DOI:** 10.3390/plants14223454

**Published:** 2025-11-12

**Authors:** Jingyu Sun, Xinyi Zhuo, Xin Chen, Jiaqi Wei, Jiali Lin, Huimin Shang, Nan Shan, Yingjin Huang, Qinghong Zhou, Zihao Li

**Affiliations:** 1Jiangxi Province Key Laboratory of Vegetable Cultivation and Utilization, Jiangxi Agricultural University, Nanchang 330045, China; sunjingyu@jxau.edu.cn (J.S.); yuanyiwenhua2020@126.com (X.Z.); chenxin24@itbb.org.cn (X.C.); 18646280831@163.com (J.W.); 13407975685@163.com (J.L.); shanghuimin12345@163.com (H.S.); shannan@jxau.edu.cn (N.S.); yjhuang_cn@126.com (Y.H.); 2Key Laboratory of Crop Physiology, Ecology and Genetic Breeding, Ministry of Education, Jiangxi Agricultural University, Nanchang 330045, China; 3Ningdu County Bureau of Agriculture and Rural Affairs, Ningdu 342899, China

**Keywords:** *D. alata*, anthocyanin, key genes, transcriptome

## Abstract

Anthocyanins can enhance the nutritional and market value of *Dioscorea alata* L. They are synthesized in a tissue-specific manner in the peel and flesh of tubers in some *Dioscorea alata* L. varieties, yet the regulatory mechanisms behind this remain unclear. In order to identify the genes involved in anthocyanin synthesis between the skin and flesh of *D. alata*, three varieties exhibiting distinct anthocyanin phenotypes were studied. A comprehensive analysis of the skin and flesh was conducted to identify the presence of anthocyanins. Three identical anthocyanins were identified in both the skin and the flesh: Alatanin C, Cya-3-*O*(2-O-glucosyl) glu, and Cya-3-*O*(6-O-sinapoyl) sop-5-O-glu. To investigate the anthocyanin biosynthesis pathways in purple *D. alata* skin and flesh, transcriptome sequencing was performed on both tissues. This analysis identified eight anthocyanins in the skin and fifteen in the flesh. Cyanidin-type anthocyanins were found to be the most abundant type of anthocyanin in both skin and flesh. Subsequent identification of 30 key genes associated with anthocyanin biosynthesis revealed *4CL* (4-coumarate CoA ligase) and *DFR* (dihydroflavonol 4-reductase) as potential key regulators of anthocyanin variation between skin and flesh. This study is of considerable theoretical and practical significance for the genetic enhancement of anthocyanin traits in *D. alata*.

## 1. Introduction

*Dioscorea alata* L. originates from south east Asia, it is the yam species most widely spread throughout the world and is the second largest food crops just less than cassava in some countries in Africa due to rich nutrition, delicious taste, and bright color [1]. *D. alata* is also a highly distinctive and nutritionally valuable vegetable crop in southern China, it is well known for its high yield, primarily attributed to its broad adaptability to both tropical and subtropical climatic conditions [2]. The tuber flesh or skin of *D. alata* exhibits a range of colors, varying from white, yellowish, orange, and pink to purple [3,4]. Anthocyanins as a crucial class of secondary metabolites belonging to the flavonoid family in plants, typically impart red, blue, or purple hues to plant tissues such as roots, leaves, flowers, fruits, and tubers [5,6]. Anthocyanins confer multiple functions to plants, including facilitating pollination, promoting seed dispersal, and enhancing stress resistance [7,8,9]. Additionally, they exhibit significant health-promoting effects in humans, particularly in the aspects of anti-aging, skin health maintenance, and cardiovascular disease prevention [10].

The biosynthesis of anthocyanins in plants constitutes a key branch of the flavonoid metabolic pathway. This biosynthetic process initiates with the conversion of phenylalanine to *p*-coumaroyl-CoA, which proceeds via three sequential enzymatic reactions [11]. In plants, anthocyanin biosynthesis involves two distinct classes of structural genes: early biosynthetic genes (EBGs) and late biosynthetic genes (LBGs) [5,12]. Specifically, EBGs—including chalcone synthase (CHS), chalcone isomerase (CHI), flavonol 3-hydroxylase (F3H), and flavonoid 3′-monooxygenase (F3′H)—are responsible for generating common precursors essential to the flavonoid biosynthesis pathway. In contrast, LBGs, such as dihydroflavonol 4-reductase (DFR), anthocyanin synthase (ANS), anthocyanidin 3-*O*-glucosyltransferase (UGT), and leucoanthocyanidin reductase (LAR), are specifically dedicated to anthocyanin biosynthesis [13,14]. Among which, silencing of DFR genes was found to affect the expression of other genes involved in the flavonoid biosynthetic pathway in floral tissues and alter flower color in tobacco plants [15]. By overexpressing the gene encoding flavonoid 3′,5′-hydroxylase (F3′5′H), carnations, roses, and chrysanthemums can also produce delphinidin and thus exhibit a blue color [16]. Excessive accumulation of the *UFGT* during the tuber development stage of Norland potatoes may inhibits anthocyanin synthesis [17].

Anthocyanin biosynthesis genes are typically regulated by MYB transcription factors (MYB TFs), basic helix-loop-helix (bHLH) proteins, and WD40 repeat proteins (WD40), as well as by the MYB/bHLH/WD40 (MBW) complex [18,19,20]. Additionally, anthocyanin biosynthesis is also transcriptionally regulated by WRKYGQY (WRKY) transcription factors [21], basic leucine zipper (bZIP) [22], and NAC domain transcription factor (NAC) [23]. These transcription factors usually regulate anthocyanin synthesis by interacting with structural genes, EBGs and LBGs. For example, StAN11 (a WD40-repeat gene) regulates anthocyanin biosynthesis in potato by controlling DFR expression [24]. In Asiatic hybrid lily (*Lilium* spp.), LhWRKY44 positively regulates anthocyanin accumulation by two mechanisms: enhancing the activating effect of the MBW complex, and binding to and activating the promoters of *LhF3H* as well as the anthocyanin-related glutathione S-transferase gene (*LhGST*) [25]. In red-fleshed apples, MdNAC1 positively regulates anthocyanin synthesis by interacting with MdbZIP23 to activate the transcription of both MdMYB10 and MdUFGT [26].

In *D. alata,* transcriptome analysis revealed the significant upregulation of the anthocyanin biosynthesis-related genes *CHS*, *F3H*, *F3′H*, *DFR*, *ANS*, and *UF3GT* in a *D. alata* cultivar with purple tuber flesh compared to a *D. alata* cultivar with white tuber flesh [27]. In addition, the expression levels of *Phenylalanine ammonia-lyase* (*PAL*), *F3H*, *ANS*, and *UGT*, encoding enzymes related to anthocyanin biosynthesis, exhibited some correlation with anthocyanin accumulation rates in various organs of *D. alata* [28]. The metabolomics analysis demonstrated that 424 metabolites, including 104 flavonoids and 8 tannins, accumulated differentially in purple tubers and white tubers [29]. We have also demonstrated through experiments that DaMYB75 and DaMYB56 antagonistically regulate anthocyanin biosynthesis by binding to the DaANS promoter in *D. alata* flesh [30]. Despite these relevant studies conducted previously, the research objects have mostly focused on the tuber flesh of *D. alata*, and no systematic and simultaneous studies have been carried out on the differences in anthocyanin synthesis between tuber flesh and tuber skin.

In this study, we used metabolomics and transcriptomics analyses to elucidate anthocyanin synthesis, regulation, and accumulation in the tuber flesh and tuber skin of different colored *D. alata* varieties. Our findings reveal targeted genes and regulators that are meaningful for improvement of *D. alata* tuber colors and also nutritional quality via molecular breeding and genetic manipulation.

## 2. Results

### 2.1. Overview of Metabolome Anthocyanins in D. alata

In order to ascertain the disparities in anthocyanins between the skin and flesh of *D. alata* tubers, the anthocyanin glycoside composition of the epidermis and flesh from mature tubers of three *D. alata* varieties (Figure 1a), purple skin with purple flesh, purple skin with white flesh, and white skin with white flesh) was analysed.

The results of the study indicated that the predominant type of anthocyanins present in the skin was found to be one of 8 varieties. The analysis yielded a total of 5 cyanidin and 3 peonidin derivatives. The flesh exhibited greater diversity, comprising 15 anthocyanins: 9 cyanidin, 1 peonidin, 2 delphinidin, 1 pelargonidin, and 2 petunidin. Furthermore, the anthocyanin glycoside content exhibited higher accumulation in the purple-skin with purple-flesh (pps/ppf) category compared to the purple-skin with white-flesh (pws), white-skin with white-flesh (wws), purple-skin with white-flesh (pwf), and white-skin with white-flesh (wwf) categories.

Subsequent PCA clustering provided a concise overview of the sample distribution, thereby highlighting the presence of discernible variations in anthocyanin composition and content. The accumulation of anthocyanins exhibited significant variation among the *D. alata* skin samples (pps, pws, and wws) (Figure 1b), while the *D. alata* flesh samples (ppf, pwf, and wwf) also demonstrated distinct disparities (Figure 1c).

### 2.2. Types of Anthocyanins of D. alata Skin and Flesh

Eight anthocyanins were identified in the skin of three *D. alata* species: Alatanin C (Cya-3-6-sin), Cya-3-O(2-O-glucosyl) glu, Cya-3-O(6-O-sinapoyl) sop-5-O-glu, Cya-3-O-glu, Cya-3-O-sop, Peo-3,5-O-dig, Peo-3-O(6-O-caffeoyl)-glu, and Peo-3-O-glu. Among these, Cya-3-O(2-O-glucosyl) glu and Cya-3-O(6-O-sinapoyl) sop-5-O-glu exhibited the highest content in pps and were identified as the primary factors responsible for the purple coloration in *D. alata* skins (Figure 2a).

Moreover, a total of fifteen anthocyanins were identified in the flesh of three *D. alata* species: Alatanin 2 (Peo-3,4-sin), Alatanin C (Cya-3,6-sin), Cya-3-(6-fer)-5-glu, Cya-3-O-(2-O-glucosyl)-glu, Cya-3-O(6-O-sinapoyl)sop-5-O-glu, Cya-3-O-gen, Cya-3-O-rut, Cya-3-O-sam, Cya-3-di-glucoside-5-glu, Cya-3-O-caffeoylglucoside-5-O-glu, Del-3-O-(6″-O-p-coumaroyl)-glu, Del-3-O-rutinoside-7-O-glu, Pel-3-O-rut, Pet. The structures of both 3-O-(6″-O-caffeoyl)-glucose and 3-O-(6″-O-p-coumaroyl)-glucose have been determined (Figure 2b). Among these, Alatanin C (Cya-3-6-sin) exhibited the highest content in ppw and was identified as the primary factor contributing to the purple coloration in *D. alata* flesh. Consequently, the presence of anthocyanins of different types and concentrations, particularly Cyanidin and its derivatives, has been demonstrated to enhance anthocyanin deposition in tuberous *D. alata* flesh (Figure 2c).

Furthermore, we identified differential metabolites among pps, pws, and wws, with a total of eight metabolites, primarily cya and peo derivatives, demonstrating higher accumulation in pps. The differential metabolites in ppf, pwf, and wwf did not align with those in the *D. alata* skin. Fifteen metabolites accumulated to a greater extent in ppf, predominantly derivatives of cya, peo, del, pel, and pet (Figure 2c). The results of this study lend support to the hypothesis that the variety and content of anthocyanins in *D. alata* flesh exceed those in the skin.

### 2.3. Screening of Differentially Expressed Genes in D. alata Skin and Flesh

In order to conduct a more in-depth investigation into the molecular mechanisms that underpin anthocyanin synthesis in the flesh and skin of *D. alata*, comprehensive transcriptome sequencing analyses were performed on pps, pws, wws, ppf, pwf, and wwf. Three replicates were collected for each sample, yielding between 43,184,984 and 43,255,612 clean reads per experimental sample, resulting in a total of 18 RNA-seq libraries (Appendix A).

Further analysis identified 550, 536 and 371 shared specific expressed among the pps, pws and wws samples, and 880, 708 and 468 shared specific expressed among the ppf, pwf and wwf (Figure 3a). Utilising a *p*-value cut-off of 0.05 and a |log2 fold change| of at least 1, a total of 10,892, 11,581, and 9208 differentially expressed genes (DEGs) were identified in the pws_vs_pps, wws_vs_pps, and wws_vs_pws samples, respectively. Among these genes, 5758, 6532, and 5084 up-regulated genes and 5134, 5049, and 4124 down-regulated genes were identified in the pws_vs_pps, wws_vs_pps, and wws_vs_pws treatment groups, respectively (Figure 3b).

In the *D. alata* flesh group, 9497, 9733, and 8910 differentially expressed genes (DEGs) were identified in the pwf_vs_ppf, wwf_vs_ppf, and wwf_vs_pwf samples, respectively. Among these genes, 5255, 5216, and 4432 up-regulated genes and 4242, 4517, and 4478 down-regulated genes were identified in the pwf_vs_ppf, wwf_vs_ppf, and wwf_vs_pwf treatment groups (Figure 3b).

In order to validate the enrichment of these GO terms across the entire transcriptome, gene set enrichment analysis (GSEA) was employed to compare pps, pws, wws, ppf, pwf, and wwf. It is noteworthy that genes implicated in the flavonoid biosynthesis pathway exhibited substantial enrichment and were distinctly overexpressed in pps and wws (Appendix A). The DEGs were then categorised into three groups via the GO database: molecular function, cellular component, and biological process (Appendix A). An analysis of differentially expressed genes in *D. alata* skin and flesh revealed that the majority were predominantly distributed in the cellular component and molecular function categories. KEGG pathway enrichment analysis revealed that, in *D. alata* skin, differentially expressed genes were primarily enriched in metabolic pathways, including Biosynthesis of secondary metabolites, Flavonoid biosynthesis, and Metabolic pathways (Figure 4). An analysis of differentially expressed genes in potato flesh revealed that the majority were predominantly enriched in metabolic pathways, including those involved in the Biosynthesis of secondary metabolites, Flavonoids, Anthocyanins and Metabolic pathways. Genes associated with anthocyanin synthesis and regulation constituted a substantial proportion within these pathways, including F3H, ANS, and UGT.

### 2.4. WGCNA Screens Genes Related to Anthocyanins in D. alata

To investigate the differences in genes involved in the anthocyanin biosynthesis pathway between skin and flesh, weighted gene co-expression network analysis (WGCNA) was applied to the DEG data and 21 anthocyanins from *D. alata* skin and flesh. The WGCNA identified six gene modules (Figure 5a, Appendix A), with the number of genes in each module ranging from 19 to 3920.

In particular, genes within the MEyellowgreen module exhibited significant positive correlations with anthocyanins, including Cya-3-O(2-O-glucosyl) glu, Cya-3-O(6-O-sinapoyl) sop-5-O-glu, Cya-3-O-glu, and Peo-3-O-glu (Figure 5b). These included genes such as CHS, F3H, and transcription factors like WRKY. Genes in the MEpaleturquoise module demonstrated significant positive correlations with anthocyanins, including Cya-3-O-sop, Peo-3,5-O-dig, and Peo-3-O-glu. These include genes such as C4H, F3`H, DFR, and transcription factors like WRKYs and bHLHs. Genes within the MElightcyan module demonstrated significant positive correlations with anthocyanins, including alatanin 2, Cya-3-(6-fer)-5-glu, Cya-3-O-gen, Cya-3-O-caffeoylglucoside-5-O-glu, Del-3-O-(6″-O-p-coumaroyl)-glu, Del-3-O-rutinoside-7-O-glu, and Pet-3-O-(6″-O-caffeoyl)-glu, including F3′5′H and UGTs. Furthermore, genes within the MEyellowgreen module exhibited significant negative correlations with anthocyanins, including Alatanin 2, Cya-3-(6-fer)-5-glu, and Cya-3-O-gen, involving F3′5′H, ANS, UGT genes, and bHLHs, WRKYs. Genes in the MEcyan module exhibited significant negative correlations with anthocyanins, including Alatanin C, Cya-3-O(2-O-glucosyl) glu, Cya-3-O-glu, and Cya-3-O-sop. These correlations involved PAL, 4CL, UGTs, and certain WRKYs.

### 2.5. Identification of Key Genes Controlling Anthocyanin Biosynthesis in D. alata Skin and Flesh

Transcriptional levels were analysed for 30 candidate genes associated with anthocyanin biosynthesis, which had been identified through the use of homology mapping with other species (Figure 6). In the context of the phenylalanine-to-naringenin biosynthetic pathway, a comprehensive analysis was conducted to elucidate the regulatory mechanisms underlying the control of anthocyanin deposition. The investigation revealed that upstream of this pathway, two *PAL* genes (PAL1, 2), one *C4H* (C4H), five *4CLs* (4CL1, 2, 3, 4, 5), four *CHS* genes (CHS1, 2, 3), and three *CHIs* (CHI1, 2, 3) exhibited significant upregulation in both ppf and pps. Notably, the expression levels of these genes were found to be higher in the flesh compared to the skin, suggesting a potential role for these genes in regulating anthocyanin deposition.

Subsequently, a comparison was made between the gene expression of naringenin and that of anthocyanin. A total of three *F3H* genes (F3H1, 2, 3), two *F3′H* genes (F3′H1, 2), four *F3′5′H* genes (F3′5′H1, 2, 3, 4), three *DFR* genes (DFR1, 2, 3), one *ANS* gene, and two *UGT* genes (UGT1, 2) were identified. The majority of these genes were found to be significantly upregulated in pp and pw compared to ww, with pp exhibiting higher expression levels than pw.

It is noteworthy that the level of *F3H* expression was found to be significantly low in pwf and wwf, while it was markedly elevated in ppf. This finding indicates the indispensable role of F3H in enhancing anthocyanin deposition in tuber flesh. *F3′H* and *F3′5′H* convert naringenin into distinct pathways, yielding red or purple pigments, respectively. Among these, *F3′H4* was found to be significantly upregulated in pp, suggesting a potential role in promoting metabolic influx into cya- and peo-related biosynthetic pathways. Furthermore, only *DFR1* exhibited a comparatively low-abundance increase in ppf, while *DFR2/3* demonstrated a significant upregulation in both pps and pws. The results obtained from this study indicate that *DFR1* exhibits a high level of acquisitiveness in anthocyanin deposition within the parenchyma of yams. Furthermore, the analysis revealed that *F3′5′H* is expressed at a significantly higher level in pp samples compared to other samples, suggesting that this enzyme may play a pivotal role in enhancing metabolic flux for the deposition of purple pigments. Concurrently, *ANS* and *UGT* levels increased significantly, promoting anthocyanin deposition in pps and ppf samples.

The results obtained from this study indicate that there were higher levels of anthocyanin pathway gene expression in ppf than in pwf and wwf, higher levels in pps than in pws and wws, and higher levels in skin than in flesh.

### 2.6. RT-qPCR Validation of Key Anthocyanin-Related Genes in Transcriptomic

To further verify the function of genes regulating anthocyanin biosynthesis, the RT-qPCR technique was used to detect the expression levels of 8 key structural genes (PAL, C4H, 4CL, CHS, F3H, F3′H, F3′5′H, and ANS) in 18 samples. The trends in RT-qPCR expression were consistent with those of the FPKM values. Further correlation analysis yielded a correlation coefficient (R^2^) of 0.8183, indicating that the transcriptome data were accurate and reliable (Figure 7). The RT-qPCR results were consistent with the transcriptome data, which further confirmed the key role of these genes in the metabolic differentiation of anthocyanins in *D. alata*.

## 3. Discussion

Anthocyanins are the most important plant natural flavonoid pigments, conferring purple, red, and blue colors to plant tissues, such as leaves, stems, roots, flowers, and fruits [31]. Among root and tuber crops, varieties with higher anthocyanin content typically possess greater nutritional value and market value [32]. In recent years, extensive research has been conducted on the anthocyanin composition and biosynthesis in root and tuber plants. In *Ipomoea batatas*, UHPLC-DAD-Orbitrap analysis of the extracts identified 18 high-confidence anthocyanin compounds, predominantly acylated peonidin and cyanidin derivatives, which accounted for over 90% of the total anthocyanin signal [33]. For Solanum tuberosum, metabolomic results indicated that the four colored potato cultivars contained abundant polyphenols, flavonoids, and anthocyanins. Cyanidin, delphinidin, and malvidin were identified as the major anthocyanidins in purple potatoes, whereas red potatoes consisted mainly of pelargonidin and its derivatives [34]. In *Raphanus sativus*, the root skin of radish contains 16 anthocyanin types, 12 of which are cyanidin and its derivatives. Notably, cyanidin 3-*O*-glucoside maintains a high concentration across different developmental stages of black radish [35]. In *Dioscorea alata*, previous studies have shown that procyanidin and cyanidin glycosides, such as cyanidin-3-*O*-(sinapoyl)sophoroside, cyanidin-3-*O*-sophoroside, procyanidin B1, procyanidin B3, and quercetin-3-*O*-glucoside, were the primary anthocyanin components [36]. In our study, 8 anthocyanins were identified in the tuber skin and Cyanidin-3-*O*(2-*O*-glucosyl) glu and Cyanidin-3-*O*(6-*O*-sinapoyl) sop-5-*O*-glu exhibited the highest content and were identified as the primary factors responsible for the purple coloration in *D. alata* skins. While a total of 15 were detected in the tuber flesh, Alatanin C (Cya-3-6-sin) exhibited the highest content in ppf and was identified as the primary factor contributing to the purple coloration in *D. alata* flesh. Therefore, it can be largely confirmed that cyanidin and its derivatives are the primary cause of the purple coloration in yam tubers. As for the differences in other anthocyanin types identified in yam tubers across different studies, these are likely attributed to variations in the plant’s growth environment, such as light, temperature, moisture, and soil type.

The integration of transcriptomic and metabolomic analysis has been demonstrated to offer a more comprehensive understanding of the mechanisms that regulate transcription within metabolic pathways. This approach has been shown to facilitate the elucidation of the processes involved in the accumulation of plant metabolites [13]. For example, an integrated metabolome and transcriptome analysis was performed to investigate the composition and content of anthocyanins in pea (*Pisum sativum* L.) pods of different colors (such as green, yellow, and purple), and five key genes (*PAL*, *4CL*, *CHS*, *F3H*, and *UFGT*) involved in anthocyanin synthesis were found to have higher expression levels in purple pea pods [37]. Combined metabolomic and transcriptomic analysis of grape (*Vitis vinifera* L.) revealed 66 flavonoids with significant differences between the flesh and skin. The main flavonoids in the flesh were pelargonidin and peonidin, while those in the skin were peonidin, delphinidin, and petunidin. Additionally, 57 structural genes related to flavonoid synthesis showed significant differences in expression: ANS2 was mainly expressed in flesh, ANS1 was expressed in the skin, and DFR had relatively high expression levels in both flesh and the skin. Furthermore, 12 ERF genes, 9 MYB genes, and 3 bHLH genes associated with this process were screened out [38]. In cassava, metabolomic and transcriptomic analyses were conducted to explore the mechanisms underlying color formation in seven developmental stages of yellow and white-rooted varieties. This analysis identified two transcription factors, MeMYB5 and MeMYB42, which were found to be co-expressed with genes involved in anthocyanin biosynthesis [39]. The present study performed transcriptome sequencing on the skin and flesh of three differently colored *D. alata*, combined with anthocyanin data from both tissues, and identified 30 key genes associated with anthocyanin biosynthesis in *D. alata*.

ANS is a 2-ketoglutarate-dependent dioxygenase that catalyses the penultimate step in anthocyanin biosynthesis, thereby converting naturally colorless anthocyanins into colored anthocyanins [40]. For instance, *CpANS1* is highly expressed in the red perianth segments of winter jasmine but scarcely detected in yellow perianth segments [13]. Li et al. enhanced anthocyanin accumulation in Salvia miltiorrhiza by overexpressing *SmANS* and altering its phenolic acid content [41]. UGT represents the final pivotal enzyme in the anthocyanin synthesis pathway, chiefly catalysing the formation of stable anthocyanins by linking unstable anthocyanins with glucuronic acid via glycosidic bonds for subsequent storage in vacuoles [42]. Wei et al. successfully introduced the UGT into potatoes via Agrobacterium co-cultivation. A detailed analysis of the resulting transgenic plants was conducted, which revealed a darker tuber coloration and a significantly elevated anthocyanin content in comparison to the wild-type plants [43]. A comparative RNA-seq analysis of purple-fleshed and yellow-fleshed sweet potatoes was conducted, which led to the identification of UF3GT as a pivotal enzyme in the anthocyanin synthesis pathway [44]. Furthermore, the analysis of the transcriptome and RT-qPCR across various tissues of both white-fleshed and purple-fleshed sweet potato varieties has provided additional evidence that structural genes such as *F3H*, *ANS*, and *UGT* play a crucial role in the formation of tuber coloration [27,28]. In this study, *DFR1* exhibited a comparatively low-abundance increase in ppf, while *DFR2/3* demonstrated a significant upregulation in both pps and pws. The results obtained from this study indicate that *DFR1* exhibits a high level of acquisitiveness in anthocyanin deposition within the parenchyma of *D. alata*.

Anthocyanins in plants often exhibit the characteristic of tissue-specific biosynthesis, which is regulated by elaborate and complex mechanisms. In genus *Actinidia* (kiwifruit), the accumulation of anthocyanins may occur in all fruit tissues, or may be restricted to the skin, peel, or a part of the peel, usually the endocarp [45]. Studies have shown that AcMYB123 and AcbHLH42 interact to form a complex, which then binds to the promoters of AcANS and AcF3GT1 thereby activating their expression and promoting anthocyanin synthesis in the inner pericarp [46]. Anthocyanins also exhibit tissue-specific accumulation in the skin and flesh of potato tubers. StMYB3 regulates anthocyanin synthesis in the tuber skin by binding to the promoters of structural genes such as *StCHS*, *StF3′5′H*, *StDFR*, and *StF3H* [47]. Within the *Pigmented tuber flesh* (*Pf*) locus, StMYB200 promotes the transcription of StMYB210 by directly binding to a 1.7 kb insertion in the StMYB210 promoter, thereby regulating anthocyanin biosynthesis and tuber flesh color in potato [48]. In the present study, Anthocyanins are specifically synthesized in the peel or flesh of *D. alata*. tubers. Regulatory network and tissue expression pattern analyses identified several transcription factors (TFs)—including MYB, WRKY, NAC, and bHLH families—that were significantly correlated with structural genes such as CHS, F3H, C4H, F3′H, DFR, F3′5′H, and UGTs, and were specifically highly expressed in either the tuber skin or flesh of *D. alata*. These TFs may participate in regulating the expression of structural genes during anthocyanin accumulation in *D. alata* tubers. However, the specific functions of these candidate genes in regulating anthocyanin biosynthesis require further experimental verification.

## 4. Materials and Methods

### 4.1. Plant Materials

The present study utilized three yam varieties developed by the Yam Germplasm Resource Innovation and Utilization Team at Jiangxi Agricultural University. The following cultivars have been identified: Gan Zi Shi 2 (pp, purple skin and purple flesh), Gan Zi Yu 3 (pw, purple skin and white flesh), and Gan Bai Yu 1 (ww, white skin and white flesh). These varieties have been observed to exhibit relatively consistent growth periods and characteristic tuber coloration. The materials were planted at the Yam Cultivation Base of Jiangxi Agricultural University (Nanchang City, 115.84 E, 28.77 N) on 8 April 2024. Standard field management practices were adopted, including appropriate fertilization, irrigation, and pest and disease control. At the rapid tuber expansion stage (150 days after planting), yam tubers were excavated. After quick cleaning, a 1 mm-thick layer was peeled from the tuber surface with a knife as the tuber skin, and the central part of the tubers was sampled separately as the tuber flesh. Three biological replicates were collected for each cultivar. All samples were rapidly frozen and stored at −80 °C in ultra-low temperature freezers for subsequent experimentation.

### 4.2. Metabolome Profile and Quantitation for Anthocyanin Metabolites

The sample was weighed out at 0.2 g, then rapidly ground into a powder using liquid nitrogen. Thereafter, 3 mL of extraction solution (acetone: distilled water: formic acid at a ratio of 80:19.8:0.2) was added to a 10 mL brown centrifuge tube. The sample was subjected to shaking at 4 °C for 30 min (overnight extraction is permissible), followed by centrifugation at 7000 rpm for 10 min. This process is then repeated on three separate occasions. Evaporation of the extraction solution to dryness at 30 °C using a rotary evaporator is the subsequent step in the process, the purpose of which is to remove acetone. The solution was diluted with distilled water until the volume was 10 mL. Subsequent parameter determination requires storage at 4 °C.

The analysis of anthocyanins was conducted by Metware Biotechnology Co., Ltd., Wuhan, China, in accordance with their standard procedure. The primary instrumentation utilized for data acquisition was ultra-performance liquid chromatography (UPLC) (SHIMADZU Nexera X2) and tandem mass spectrometry (MS/MS) (Applied Biosystems 4500 QTRAP, Boston, MA, USA). Liquid Chromatography Conditions: Column: The Agilent SB-C18 1.8 µm, 2.1 mm × 100 mm, is utilized as the mobile phase. Phase A was ultrapure water (supplemented with 0.1% formic acid), Phase B was acetonitrile (supplemented with 0.1% formic acid); Elution gradient: The second phase commenced at 5% at 0.00 min, and increased linearly to 95% over 9.00 min. The 95% level was then sustained for 1 min. From 10.00 to 11.10 min, the B ratio decreased to 5%, and then equilibrated at this level until 14 min. The flow rate was 0.35 mL/min, the column temperature was 40 °C, and the injection volume was 4 μL. The apparatus is connected to a linear ion trap mass spectrometer (LIT) and triple quadrupole (QQQ) scanning. The offline raw data were then opened and browsed using Software Analyst 1.6.3 for qualitative and quantitative analysis. The horizontal axis represented the retention time (Rt) of metabolite detection, while the vertical axis showed the ion current intensity (cps, counts per second) of ion detection.

The metabolite content data were then normalised using the range method in R software 4.2.1 (www.r-project.org/). Hierarchical cluster analysis (HCA) was then performed to analyse the accumulation patterns of metabolites across different samples. The differential metabolites were screened by combining the fold change and VIP values from the OPLS-DA model, and annotated using the KEGG database.

### 4.3. Transcriptome Sequencing and Screening for Differential Gene Expression

The RNA-Seq was carried out by Wuhan Maiwei Metabolic Biotechnology Co., Ltd. (Wuhan, China). Following the successful extraction and quality control of the RNA, the construction of cDNA libraries was conducted in accordance with the NEBNext^®^ Ultra™ RNA Library Prep Kit for Illumina^®^ (NEB, Ipswich, MA, USA). Subsequent to library quality control, sequencing was performed on the Illumina HiSeq novaseq 6000 platform (Illumina, Inc., San Diego, CA, USA) using flow cells.

FPKM (Fragments Per Kilobase of exon model per Million mapped fragments) was utilized as the metric for gene expression levels. Genes exhibiting a log2Fold Change of at least 1 and a false discovery rate (FDR) of less than 0.05 were designated as differentially expressed genes (DEGs) [49]. Principal component analysis (PCA) was performed on the transcriptome data. The raw files of the fluorescence images generated by the Illumina platform were converted into short reads by means of base calling. These short reads were stored in the FASTQ format. The clean reads were in alignment with the reference genome, followed by analysis using DESeq2 (version 1.36.0). We applied *p* < 0.05 and minimum fold change |log2(fold change)| ≥ 1 to identify significant differences between treated and control samples. Enrichment terms for DEGs in the GO and KEGG databases showing significant differences were adjusted using TBtools v2.356 with a *p* value (FDR) < 0.05. The classification of genes was undertaken utilising the KEGG database (https://www.kegg.jp/, accessed on 26 March 2025) [50,51], the Nr database (NCBI Non-Redundant Protein Sequences) [52], the Nt database (NCBI Non-Redundant Nucleotide Sequences), the Pfam database (http://pfam.xfam.org/), and the Swiss-Prot database (https://www.uniprot.org/) [53].

### 4.4. Weighted Correlation Network Analysis

In light of the aforementioned findings, the differential expression data was integrated into WGCNA [54]. The analysis was conducted utilizing the WGCNA package in R software. The automatic network construction function was utilized to obtain co-expression modules with default parameters, with the exception of a soft threshold power. The unsigned networks were detected using the Pearson method, and a topological overlap measure (TOM) was determined for each gene pair. Hierarchical clustering was performed using mean link age based on the dissimilarity of TOM. Consequently, a dendrogram was formulated, and the size of the minimum gene module was established. The steroid saponin content of the samples was utilized as phenotypic data to calculate Pearson correlations between each gene module and the various steroid saponins, as well as between the various treated samples. Following the identification of the gene modules that exhibited a significant correlation with the sterol saponin profile, inter-gene expression correlation coefficients greater than 0.1 (*p* ≤ 1 × 10^−6^) were selected as potential candidate genes for further investigation.

### 4.5. RT-qPCR

The verification analysis of the seven differentially expressed genes (DEGs) related to the biosynthesis of polysaccharide and saponin was validated by quantitative real-time PCR (RT-qPCR) with three replicates to verify the RNA-Seq data [55]. The first-strand cDNA was synthesized from total RNA using the TransScript All-in One First-Stand cDNA Synthesis Super Mix for qPCR (One-Step gDNA Removal) according to the manufacturer’s instructions. The RT-qPCR primers were designed for the seven genes using the Primer Quest Tool. The reaction was performed on a Bio-Rad CFX Connect Real-Time PCR Detection System (Bio-Rad, Berkeley, CA, USA) using the following program: The temperature was increased to 95 °C for 15 s, then maintained at this temperature for 10 min. This was followed by 40 cycles of 95 °C for 60 s. The DaEF-1a gene was selected as an internal reference gene for the normalization process, and three technical replicates were performed for each sample. The relative expression levels for each unigene were calculated using 2^−ΔΔCt^ differences. Primers are listed in Appendix A.

### 4.6. Statistical Analysis

The statistical analysis was performed and the graphs were generated using the R project (https://www.r-project.org/) [54]. The correlation between the steroidal saponin content and gene expression among different samples was explored using Spearman analysis. The statistical software SPSS V 27 was utilized to conduct a one-way analysis of variance (ANOVA) to examine metabolic parameters. The least significant difference test was employed to calculate the comparisons of mean values (*p* < 0.05).

## 5. Conclusions

The study investigated three sweet potato varieties exhibiting distinct anthocyanin phenotypes. A total of eight anthocyanins were identified in the skin, while fifteen anthocyanins were identified in the flesh. Of these, Cya-type anthocyanins constituted the highest proportion. While the flesh exhibited a higher diversity of anthocyanins in comparison to the skin, the total anthocyanin content present in the skin surpassed that observed in the flesh. Three identical anthocyanins were identified in both the skin and the flesh: Alatanin C, Cya-3-*O*(2-O-glucosyl) glu, and Cya-3-*O*(6-O-sinapoyl) sop-5-O-glu. To investigate the anthocyanin biosynthesis pathways in purple *D. alata* skin and flesh, transcriptome sequencing was performed on both tissues. Combined with anthocyanin profiling, weighted gene co-expression network analysis (WGCNA) identified 30 key genes in anthocyanin biosynthesis pathways. Among these, *4CL1* and *DFR3* exhibited higher levels of expression in the skin, suggesting that these genes may underlie the differences in anthocyanin content between skin and flesh. In conclusion, this study contributes to the theoretical understanding of anthocyanin biosynthesis in purple *D. alata*, providing a scientific basis for future breeding efforts targeting purple-fleshed varieties.

## Figures and Tables

**Figure 1 plants-14-03454-f001:**
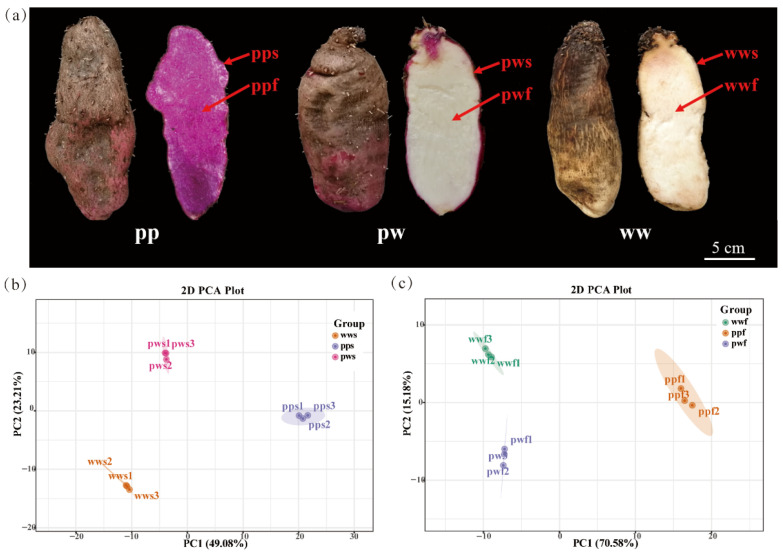
Morphology of differently pigmented *D. alata* and principal component analysis (PCA) analysis based on metabolite contents. (**a**) *D. alata* color, pp, purple skin with purple flesh, pw, purple skin with white flesh, ww, white skin with white flesh, s, skin, f, flesh. (**b**) Clustering of pps, pws, and wws samples based on anthocyanin contents. (**c**) Clustering of ppf, pwf, and wwf samples based on anthocyanin contents.

**Figure 2 plants-14-03454-f002:**
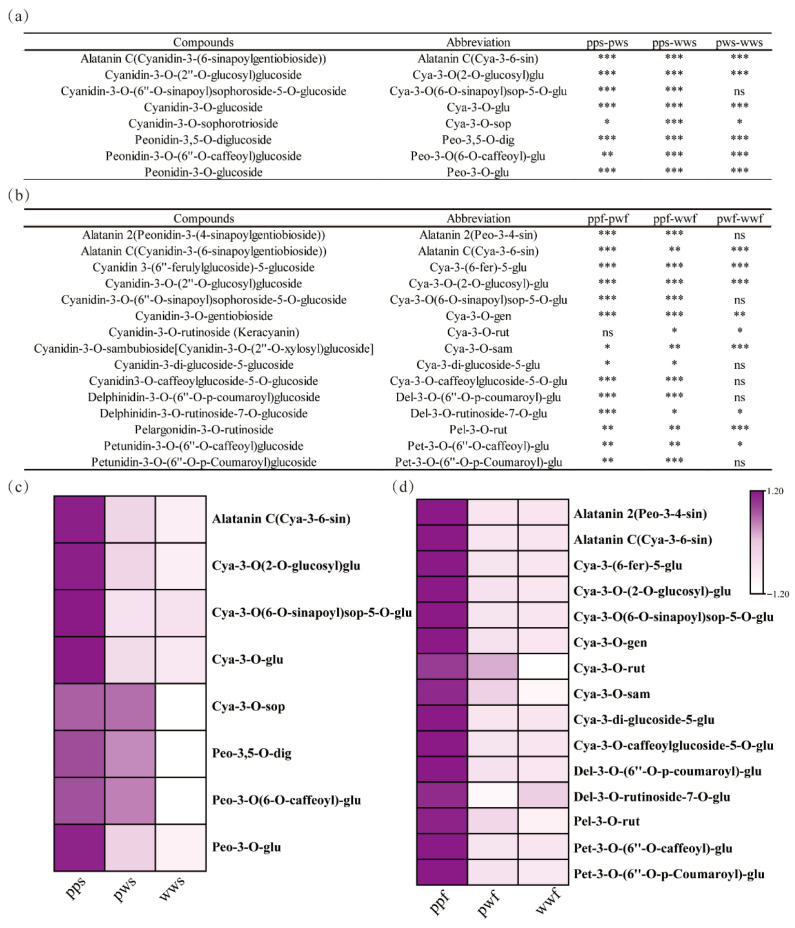
Identification of DAMs among three differently pigmented *D. alata* accessions. (**a**) Varieties and significant differences in anthocyanins among pps, pws, and wws. (**b**) Varieties and significant differences in anthocyanins among ppf, pwf, and wwf. ***, *p* < 0.001; **, *p* < 0.01; *, *p* < 0.05. (**c**) Heatmap of anthocyanins content among pps, pws, and wws. (**d**) Heatmap of anthocyanins content among ppf, pwf, and wwf.

**Figure 3 plants-14-03454-f003:**
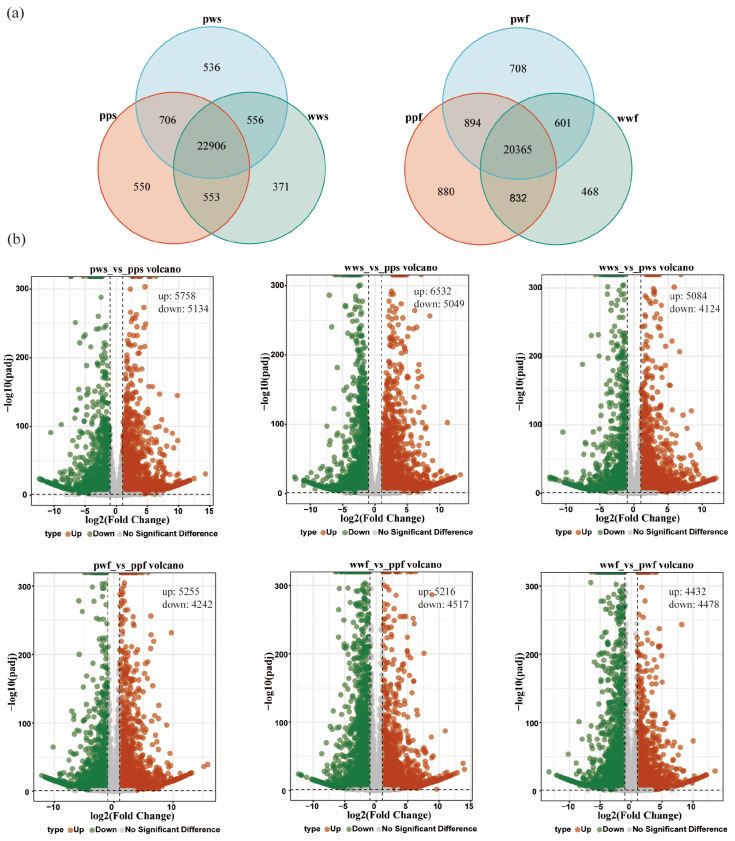
Comparative transcriptome analysis. (**a**) Venn diagram of DEGs among the samples. (**b**) DEGs of the samples. Each data point on the volcano plot corresponds to a specific gene. The x-axis displays the log_2_ (Fold Change), defined as the logarithm (base 2) of the fold change in gene expression between the two samples. The y-axis displays either the *p*-value (in cases where the number of differentially expressed genes is too small) or the log10(padj) value multiplied by −1, where padj is the probability-adjusted *p*-value.

**Figure 4 plants-14-03454-f004:**
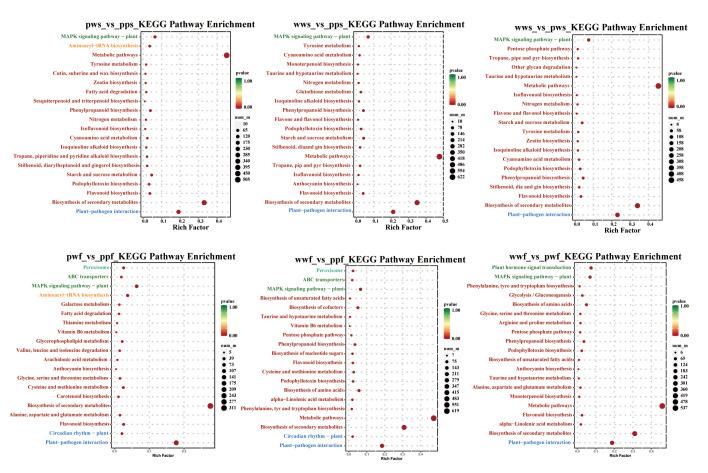
KEGG enrichment of commonly shared DEGs. The horizontal axis of the figure represents GeneRatio, defined as the proportion of differentially expressed genes in this pathway relative to the total number of differentially expressed genes. The vertical axis of the figure represents the KEGG Pathways. The color of each point denotes the *p*-value, with larger points indicating a higher number of differentially expressed genes.

**Figure 5 plants-14-03454-f005:**
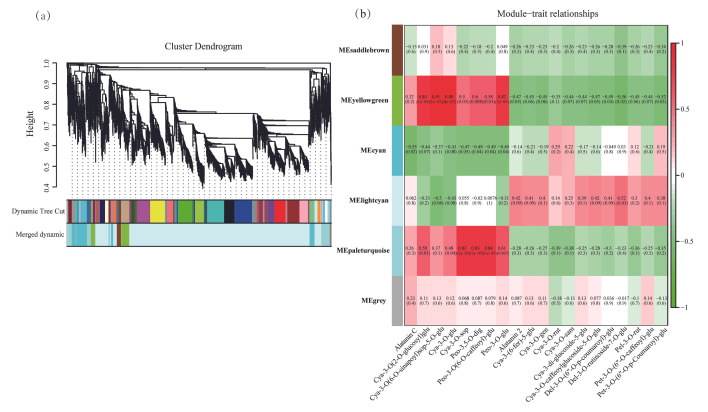
Gene co-expression network analysis among different *D. alata*. (**a**) Hierarchical clustering tree showing 6 modules of coexpressed genes by WGCNA. (**b**) Module–trait relationships. Each column represents a module, and each row represents a type of anthocyanin.

**Figure 6 plants-14-03454-f006:**
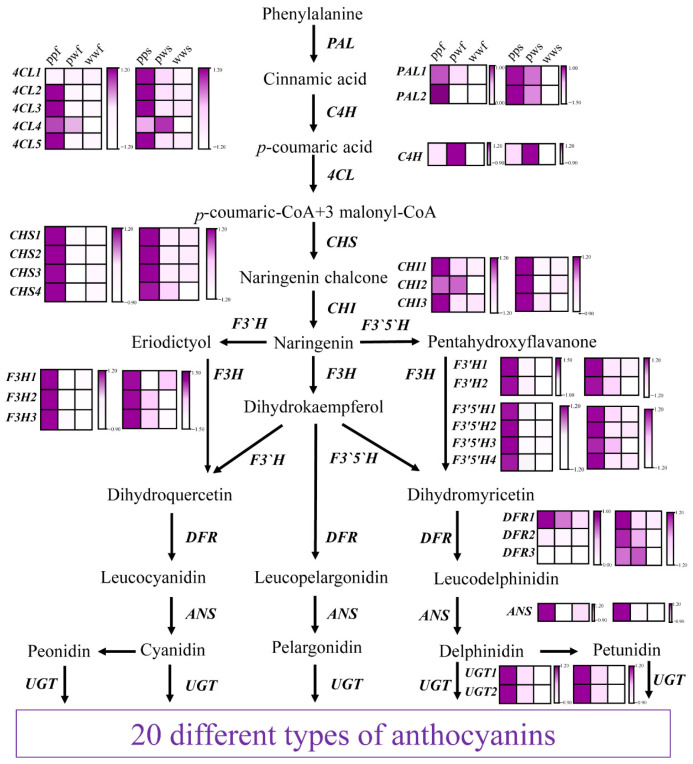
Expression of structural genes in the anthocyanin biosynthesis pathway. The block represents the amount of gene expression. PAL, phenylalaninammonialyase; C4H, cinnamate 4-hydroxylase; 4CL, 4-coumarate: CoA ligase; CHS, chalcone synthase; CHI, chalcone isomerase; F3H, flavonoid 3-hydroxylase; F3′H, flavonoid 3′-hydroxylase; F3′5′H, flavonoid 3′5′-hydroxylase; DFR, dihydroflavonol 4-reductase; ANS, anthocyanidin synthase; UFGT, UDP-glycose flavonoid glycosyltransferase. Bar: log_2_ FPKM. FPKM, fragments per kilobase of transcript per million fragments mapped.

**Figure 7 plants-14-03454-f007:**
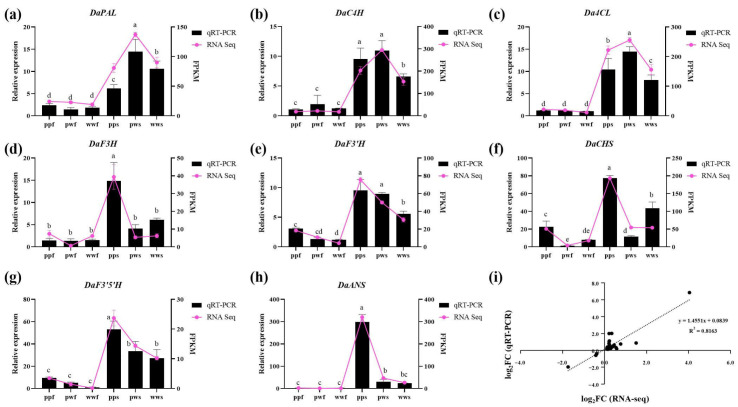
RT-qPCR of the key anthocyanin-related genes. (**a**–**h**) Expression levels of 8 genes in RT-qPCR and FPKM values in RNA-seq. (**i**) Correlation analysis of the gene expression ratios between RT-qPCR and RNA-seq. Different letters denote significant differences according to the Tukey’s test (*p* < 0.05).

## Data Availability

The original contributions presented in this study are included in the article/Appendix A. Further inquiries can be directed to the corresponding author.

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
