# Peer review of "Metabolome and Transcriptome Reveal the Regulatory Mechanism of Anthocyanin Synthesis in Tuber Skin and Flesh of *Dioscorea alata* L."

_plants, 2025, doi:10.3390/plants14223454_

Round 1
Reviewer 1 Report
Comments and Suggestions for Authors
This study analyzes the anthocyanin synthesis genes predicted to be related to organ-specific anthocyanin accumulation in the skin and flesh of yam (Dioscorea alata L.) using metabolomic and transcriptomic analyses. This work is positioned as fundamental research for D. alata, for which research examples are scarce.
However, unfortunately, the content of the Discussion is poor. It primarily consists of introductions to anthocyanin research in distantly related plant species or mere introductions of techniques. There is almost no discussion relating directly to the findings of the present study. Even if research on yam is limited, abundant studies exist on related crops like potatoes and sweet potatoes, necessitating a deeper discussion for the paper to be academically sound. Specifically, despite the Abstract and Conclusion stating that 4CL is an important regulatory factor, it is not mentioned at all in the Discussion section. The Discussion must be significantly revised.
Other comments
- Line 48: The 'p' in p-coumaroyl-CoA should be italicized.
- Line 54: The 'O' in enzyme names such as anthocyanidin 3-O-glucosyltransferase should be italicized.
- Line 55: Anthocyanin reductase (ANR) may not be involved in anthocyanin biosynthesis but rather in anthocyanin degradation (or competing proanthocyanidin synthesis).
- Line 64: Regarding the term WD repeat proteins (WD), should this be WD40 repeat proteins (WD40)?.
- Line 190: The text mentions "ginseng peel". Is this different from the usual "skin"? It needs clarification whether skin and peel refer to different parts, as both terms appear throughout the manuscript.
- Line 256: The notations qRT-PCR and RT-qPCR are mixed/inconsistent. RT-qPCR may be the more precise term.
- Line 272: Is pelargonidin-O-cinnamate correct? It is quite rare for an organic acid to bind directly to anthocyanin without a sugar intermediate, so the structure needs confirmation.
- Line 272: If the anomeric position and optical isomerism are specified, such as in malvidin-3-O-β-D-glucoside, then the notation for all other anthocyanins must also be unified.
- Line 275: References 42 and 43 concern red radish (Raphanus sativus), not carrot as stated in the text. The citation needs to be checked and corrected.
- Line 276: Geraniidin and paeoniflorin are not anthocyanins. Listing them in parallel with anthocyanins (like petunidin, delphinidin, and malvidin derivatives) is misleading and may cause confusion.
Author Response
This study analyzes the anthocyanin synthesis genes predicted to be related to organ-specific anthocyanin accumulation in the skin and flesh of yam (Dioscorea alata L.) using metabolomic and transcriptomic analyses. This work is positioned as fundamental research for D. alata, for which research examples are scarce.
However, unfortunately, the content of the Discussion is poor. It primarily consists of introductions to anthocyanin research in distantly related plant species or mere introductions of techniques. There is almost no discussion relating directly to the findings of the present study. Even if research on yam is limited, abundant studies exist on related crops like potatoes and sweet potatoes, necessitating a deeper discussion for the paper to be academically sound. Specifically, despite the Abstract and Conclusion stating that 4CL is an important regulatory factor, it is not mentioned at all in the Discussion section. The Discussion must be significantly revised.
Response: Thanks for your valuable suggestions and comments. We have carefully replied these comments and suggestions point-to-point, and revised the manuscript accordingly.
Other comments
1. Line 48: The 'p' in p-coumaroyl-CoA should be italicized.
Response: Thanks for your valuable suggestions and comments. We are sorry for this. We have changed the p in p-coumaroyl-CoA to italics in line 48.
2. Line 54: The 'O' in enzyme names such as anthocyanidin 3-O-glucosyltransferase should be italicized.
Response: Thanks for your valuable comments. We are sorry for this. We have changed the O in anthocyanidin 3-O-glucosyltransferase to italics in line 54.
3. Line 55: Anthocyanin reductase (ANR) may not be involved in anthocyanin biosynthesis but rather in anthocyanin degradation (or competing proanthocyanidin synthesis).
Response: Thanks for your valuable comments. ANR is an enzyme involved in proanthocyanidin synthesis, and we have deleted it.
4. Line 64: Regarding the term WD repeat proteins (WD), should this be WD40 repeat proteins (WD40)?
Response: Thanks for your valuable comments. WD is WD40 and we have changed WD to WD40 in line 64.
5. Line 190: The text mentions "ginseng peel". Is this different from the usual "skin"? It needs clarification whether skin and peel refer to different parts, as both terms appear throughout the manuscript.
Response: Thanks for your valuable comments. We are sorry for this. In this study, ginseng peel and skin are considered equivalent, and we have standardized the terminology to “skin” for both.
6. Line 256: The notations qRT-PCR and RT-qPCR are mixed/inconsistent. RT-qPCR may be the more precise term.
Response: Thanks for your valuable comments. We have changed qRT-PCR to RT-qPCR.
7. Line 272: Is pelargonidin-O-cinnamate correct? It is quite rare for an organic acid to bind directly to anthocyanin without a sugar intermediate, so the structure needs confirmation.
Response: Thanks for your valuable comments. We have further confirmed the relevant information on pelargonidin-O-cinnamate and revised the discussion.
8. Line 272: If the anomeric position and optical isomerism are specified, such as in malvidin-3-O-β-D-glucoside, then the notation for all other anthocyanins must also be unified.
Response: Thanks for your valuable comments. We have unified the notation for all anthocyanins.
9. Line 275: References 42 and 43 concern red radish (Raphanus sativus), not carrot as stated in the text. The citation needs to be checked and corrected.
Response: Thanks for your valuable comments. We are sorry for this. We have changed carrot to Raphanus sativus in the text in line 296.
10. Line 276: Geraniidin and paeoniflorin are not anthocyanins. Listing them in parallel with anthocyanins (like petunidin, delphinidin, and malvidin derivatives) is misleading and may cause confusion.
Response: Thanks for your valuable comments. We are sorry for this. We have delated geraniidin and paeoniflorin in the text.
Reviewer 2 Report
Comments and Suggestions for Authors
Authors investigated the D. alata cultivars of differentially colored by anthocyanin accumulation. I think this could improve the knowledge on tuber biology, yet some concerns are dragging this reviewer as following.
- Line 175~187. Please consider the k-clustering of DEGs. Simple DEG analysis seems to be less effective in this problem. Or, at least try GSEA analysis on several anthocyanin related terms. Please refer to a paper with very similar topic, [https://doi.org/10.3390/ijms23073681].
- Line 188-209. WGCNA originally aims to reveal the transcriptomic pathway underlying multiple samples. Yet, the title does not match with the explanations provided. Rather, Please provide the most related module structure and which genes are involved (WRKY, UGT and PALs).
- Line 215, It is weird term of 'structural genes'. What was the intended meaning?
- Section 2.6. It seems the qPCR result and RNA-seq result agrees yet is it possible to draw a correlation plot to know the both experiment correlates well?
- For all the figures, please reinforce the legends which lacks too much details.
- Discussion need to be re-written. Especially, discussion on the transcriptome itself is quite old and the author's work has so many other interesting features to be discussed. Please re-write to fill the biological relevance.
- Line 405, How com Bio-Rad, USA, and Wuhan China can be written in the same line? Though the product can be made from Wuhan factory, you may write a country. Please check the journal's policy and revise it.
Author Response
Reviewer 2
Authors investigated the D. alata cultivars of differentially colored by anthocyanin accumulation. I think this could improve the knowledge on tuber biology, yet some concerns are dragging this reviewer as following.
Response: Thanks for your valuable suggestions and comments. We have carefully replied these comments and suggestions point-to-point, and revised the manuscript accordingly.
1. Line 175~187. Please consider the k-clustering of DEGs. Simple DEG analysis seems to be less effective in this problem. Or, at least try GSEA analysis on several anthocyanin related terms. Please refer to a paper with very similar topic, [https://doi.org/10.3390/ijms23073681].
Response: Thanks for your valuable comments. We are sorry for this. We have added the results of the relevant GSEA analysis and placed the image in Figure S1 in line 185.
2. Line 188-209. WGCNA originally aims to reveal the transcriptomic pathway underlying multiple samples. Yet, the title does not match with the explanations provided. Rather, Please provide the most related module structure and which genes are involved (WRKY, UGT and PALs).
Response: Thanks for your valuable comments. We are sorry for this. We have revised the section titles for the WGCNA results and uploaded the gene modules identified by the WGCNA analysis to the Table S3 which genes are involved (WRKY, UGT and PALs).
3. Line 215, It is weird term of 'structural genes'. What was the intended meaning?
Response: Thanks for your valuable comments. We are sorry for this. We have changed structural genes to genes.
4. Section 2.6. It seems the qPCR result and RNA-seq result agrees yet is it possible to draw a correlation plot to know the both experiment correlates well?
Response: Thanks for your valuable suggestions. We have added a correlation plot about the qPCR result and RNA-seq result in Figure 7.
5. For all the figures, please reinforce the legends which lacks too much details.
Response: Thanks for your valuable suggestions. We are sorry for this. We have added details in captions to all the figures.
6. Discussion need to be re-written. Especially, discussion on the transcriptome itself is quite old and the author's work has so many other interesting features to be discussed. Please re-write to fill the biological relevance.
Response: Thanks for your valuable comments. We are sorry for this. We have re-written the discussion.
7. Line 405, How com Bio-Rad, USA, and Wuhan China can be written in the same line? Though the product can be made from Wuhan factory, you may write a country. Please check the journal's policy and revise it.
Response: Thanks for your valuable suggestions. We are sorry for this. We have revised the relevant information in line 462.
Reviewer 3 Report
Comments and Suggestions for Authors
16.10.2025
The manuscript entitled "Integrated Metabolomic and Transcriptomic Analyses Uncover the Molecular Mechanisms Driving Anthocyanin Variations in Tuber Skin Versus Flesh of Dioscorea alata L.” was reviewed.
The manuscript delivers novel DEGs for anthocyanin between skin and flesh of D. alata. However, the discussion was too short and weak, despite novel findings. The results are missing list of major DEGs and their fold changes. I was expected to see some insights comparing promoters of DEGs between skin and tuber flesh. Therefore, I do not recommend the publication of this manuscript in "Plants" unless these corrections are made. Please see below additional comments.
General:
- The language is in a big shift from line to line. please stick to one English style; either British or American! For example, you have British “colour” and its derivatives, which were found 10 times, while American “color” and its derivatives were listed 15 times.
- The title is too long, try to shorten it.
- Abstract:
- Lines 14-18: too long introduction please shorten to one line.
- Line 26: you need to write the genes “4CL” and “DFR” in full.
- Add some substantial findings including fold changes of major genes and regulators.
- Introduction:
- Good and informative.
- Lines 41: remove “ecological”.
- Lines 92-92: when presenting your aim at the end of the introduction, do not include any findings or results.
- Results:
- Well presented.
- Figure 3.A: at what fold DEGs are presented in the venn diagrams.
- Figure 3.B: add the number of down and up genes. Add “-regulated” to both in one word.
- Line 161-173: I expected to see a list of top 10-20 DEGs along with their fold changes for both fruit groups.
- Discussion:
- Lines 266-280: there is no discussion in theses 15 lines, please remove.
- Line 281-287: again these line are void of any discussion! Please just remove.
- The remaining paragraph is VERY Short and weak, you have novel findings that need to be discussed
- Materials and Methods:
- Line 326: please stick to one English style; either British or American! Please replace “Utilization” with “Utilisation” as the entire manuscript is written in British English.
- Line 330: please stick to one English style; either British or American! Please replace “coloration” with “colouration” as the entire manuscript is written in British English.
- Line 330: please indicate growth conditions (for reproducibility) including soil type and texture, fertilization and irrigation, growth period, harvesting date, … etc.
- Lines 336-345: please do NOT use “imperative” verbs, but rather “passive voice”, e.g. replace “should be weighed” with “it was weighed” … etc.
- Line 372: what was the read length? And were the reads paired or not?
- Line 374: a major bioinformatics section is missing, including read handling, trimming, pairing, and mapping to reference. What software or code did you apply along what conditions? What reference did you map reads to?
- Conclusion:
- It contains more data than the abstract! Add some to the abstract!
- Patents:
- Nothing is listed here.
- References:
- You need to update the reference list (24 out of 55) as 43% are published in the last five years.
- Resent highly related articles need to be added and discussed, e.g.
Zhang, P., Xu, S., Zhang, L., Li, X., Qi, J., Weng, L., ... & Wang, J. (2025). Metabolome and transcriptome profiling reveals light-induced anthocyanin biosynthesis and anthocyanin-related key transcription factors in Yam (Dioscorea Alata L.). BMC Plant Biology, 25(1), 729.
Comments on the Quality of English Language
- The language is in a big shift from line to line. Please stick to one English style: either British or American. For example, you have British “colour” and its derivatives, which were found 10 times, while American “color” and its derivatives were listed 15 times.
Author Response
The manuscript entitled "Integrated Metabolomic and Transcriptomic Analyses Uncover the Molecular Mechanisms Driving Anthocyanin Variations in Tuber Skin Versus Flesh of Dioscorea alata L.” was reviewed.
The manuscript delivers novel DEGs for anthocyanin between skin and flesh of D. alata. However, the discussion was too short and weak, despite novel findings. The results are missing list of major DEGs and their fold changes. I was expected to see some insights comparing promoters of DEGs between skin and tuber flesh. Therefore, I do not recommend the publication of this manuscript in "Plants" unless these corrections are made. Please see below additional comments.
Response: Thanks for your valuable suggestions and comments. We have carefully replied these comments and suggestions point-to-point, and revised the manuscript accordingly.
- General:
- The language is in a big shift from line to line. please stick to one English style; either British or American! For example, you have British “colour” and its derivatives, which were found 10 times, while American “color” and its derivatives were listed 15 times.
Response: Thanks for your valuable comments. We are sorry for this. We have changed the colour to color.
- The title is too long, try to shorten it.
Response: Thanks for your valuable suggestions. We have shortened the length of the title in line 2.
1. Abstract:
- Lines 14-18: too long introduction please shorten to one line.
- Line 26: you need to write the genes “4CL” and “DFR” in full.
- Add some substantial findings including fold changes of major genes and regulators.
Response: Thanks for your valuable comments. We are sorry for this. We have rewritten the abstract section and added some content.
2. Introduction:
- Good and informative.
- Lines 41: remove “ecological”.
Response: Thanks for your valuable comments. We have removed “ecological”.
- Lines 92-92: when presenting your aim at the end of the introduction, do not include any findings or results.
Response: Thanks for your valuable comments. We have re-examined the introduction section and have not included any findings or results.
3. Results:
- Well presented.
- Figure 3.A: at what fold DEGs are presented in the venn diagrams.
Response: Thanks for your valuable comments. We have added the relevant information in the figure caption for Figure 3.
- Figure 3.B: add the number of down and up genes. Add “-regulated” to both in one word.
Response: Thanks for your valuable comments. We have added the number of up and down-regulated genes in the figure 3.
- Line 161-173: I expected to see a list of top 10-20 DEGs along with their fold changes for both fruit groups.
Response: Thanks for your valuable comments. We have listed the filtered genes associated with anthocyanins and placed other filtered genes and transcription factors in Table S3.
4. Discussion:
- Lines 266-280: there is no discussion in theses 15 lines, please remove.
- Line 281-287: again these line are void of any discussion! Please just remove.
- The remaining paragraph is VERY Short and weak, you have novel findings that need to be discussed.
Response: Thanks for your valuable comments. We are sorry for this. We have rewritten the part of discussion.
5. Materials and Methods:
- Line 326: please stick to one English style; either British or American! Please replace “Utilization” with “Utilisation” as the entire manuscript is written in British English.
Response: Thanks for your valuable comments. We are sorry for this. We have changed the utilisation to utilization.
- Line 330: please stick to one English style; either British or American! Please replace “coloration” with “colouration” as the entire manuscript is written in British English.
Response: Thanks for your valuable comments. We are sorry for this. We have changed the colouration to coloration.
- Line 330: please indicate growth conditions (for reproducibility) including soil type and texture, fertilization and irrigation, growth period, harvesting date, … etc.
Response: Thanks for your valuable comments. We have added the growth conditions including soil type and texture, fertilization and irrigation, growth period, harvesting date, … etc in line 378.
- Lines 336-345: please do NOT use “imperative” verbs, but rather “passive voice”, e.g. replace “should be weighed” with “it was weighed” … etc.
Response: Thanks for your valuable comments. We are sorry for this. We have revised the relevant language.
- Line 372: what was the read length? And were the reads paired or not?
Response: Thanks for your valuable comments. We have supplemented the information from the reads in Table S2.
- Line 374: a major bioinformatics section is missing, including read handling, trimming, pairing, and mapping to reference. What software or code did you apply along what conditions? What reference did you map reads to?
Response: Thanks for your valuable comments. We have expanded the section on transcriptomic analysis and added several methods.
6. Conclusion:
- It contains more data than the abstract! Add some to the abstract!
Response: Thanks for your valuable comments. We are sorry for this. We have added some data in abstract.
7. Patents:
- Nothing is listed here.
8. References:
- You need to update the reference list (24 out of 55) as 43% are published in the last five years.
- Resent highly related articles need to be added and discussed, e.g.
Zhang, P., Xu, S., Zhang, L., Li, X., Qi, J., Weng, L., ... & Wang, J. (2025). Metabolome and transcriptome profiling reveals light-induced anthocyanin biosynthesis and anthocyanin-related key transcription factors in Yam (Dioscorea Alata L.). BMC Plant Biology, 25(1), 729.
Response: Thanks for your valuable comments. We are sorry for this. We have updated the reference list and added the relevant discussion.
Round 2
Reviewer 2 Report
Comments and Suggestions for Authors
Authors addressed some of the reviewer's comments yet not fully satisfactory.
- If you want to include the discussion on the combination of transcriptome and metabolome which is quite common in this era, you need to justify. e.g. add more examples of successful cases for the similar topics or originality of your approach compared with the others.
- Only providing gene list for the WGCNA modules even without any annotation is not useful. The reviewer suggested was deeper elaboration on the specific modules or at least some functional relevance such as GO result or so.
Author Response
- If you want to include the discussion on the combination of transcriptome and metabolome which is quite common in this era, you need to justify. e.g. add more examples of successful cases for the similar topics or originality of your approach compared with the others.
Response: Thanks for your valuable suggestions. We have added more references and discussions related to the integrated metabolomic and transcriptomic analysis in plant anthocyanin research, see lines 324-337
- Only providing gene list for the WGCNA modules even without any annotation is not useful. The reviewer suggested was deeper elaboration on the specific modules or at least some functional relevance such as GO result or so.
Response: Thanks for your valuable suggestions and comments. The genome sequence of D. alata has been completed and annotated. A comprehensive annotation file (Table S4) for the entire genome will be provided to facilitate review by editors and readers.
Round 3
Reviewer 2 Report
Comments and Suggestions for Authors
Table S4 contains whole annotation of the genomic information, which cannot be the answer for the comment as following: "Only providing gene list for the WGCNA modules even without any annotation is not useful. The reviewer suggested was deeper elaboration on the specific modules or at least some functional relevance such as GO result or so. "
The most appropriate answer would be at least discussing which module is physiologically relevant with their observation and how those genes are correlated (such as assessing GO analysis on the genes of specific module), not just uploading a table and answer as 'do it yourself'.
Also, the lines picked by the authors were not correlated with the question as well.
Thus, I personally thought these authors would not sincerely address the reviewer's comment and meet the scientific standards of the Plants.
Unfortunately, the authors are not willing to revise the manuscript enough to provide the essential insights and meanings from their results. This reviewer regret to inform the authors a rejection.